# The Evolution of Open Space Planning within a Developing, Biodiverse City (Durban, South Africa)

Cameron T. McLean [1,2,*], Debra C. Roberts [1,3,4] and Rob Slotow [5,6]

1   School of Life Sciences, University of Kwazulu-Natal, Pvt Bag X01, Scottsville,
    Pietermaritzburg 3209, South Africa; debra.roberts@palebludot.co.za
2   Biodiversity Management Department, eThekwini Municipality, Durban 4001, South Africa
3   Sustainable and Resilient City Initiatives Unit, eThekwini Municipality, Durban 4001, South Africa
4   Faculty of Geo-Information Science and Earth Observation, University of Twente,
    7500 Enschede, The Netherlands
5   Centre for Functional Biodiversity, School of Life Sciences, University of Kwazulu-Natal, Pvt Bag X01,
    Scottsville, Pietermaritzburg 3209, South Africa; slotow@ukzn.ac.za
6   Department of Genetics, Evolution & Environment, University College London, Darwin Building,
    Gower Street, London WC1E 6BT, UK
*   Correspondence: cameron.mclean@durban.gov.za

**Abstract:** Conserving and restoring biodiversity is central to the achievement of the Sustainable Development Goals. The need to curb biodiversity loss through the mainstreaming of biodiversity considerations within land-use planning is consistently highlighted in global biodiversity assessments intended for policymakers and practitioners. We present a Global South local government-led examination of the mainstreaming of biodiversity issues within a biodiversity hotspot area. Here, we evaluated the four-decade-long evolution in open space planning in Durban, South Africa, in response to shifting urbanisation, governance and policy/legislative contexts. We assessed the role of science in responding to contextual changes, the need for champions, and key institutional interventions undertaken to embed a biodiversity function within local government. In addition, we investigated how biodiversity concerns have been incorporated into land-use planning applications via the city's environmental planning function. We provide evidence of the advancement of mainstreaming biodiversity concerns within local government processes, institutional functions, and land-use decision-making. This has been achieved through effective and sustained leadership; the use of science and scientific information in advancing the policy and legislative environment and building political support by responding to shifting governance contexts; investment in institutional scientific capacity and generating scale-appropriate biodiversity information. Learnings from this paper may be useful for other local governments addressing biodiversity loss through land-use planning processes, by identifying critical investment areas that may shorten the time required for effective mainstreaming.

**Keywords:** biodiversity mainstreaming; urban biodiversity; local government; land-use planning; eThekwini

## 1. Introduction

The process of urbanisation is a defining mega trend of the 21st century [1], and over half of the global population already live in cities, expected to rise to 68% by 2050, growing mostly in the continents of Africa and Asia [2]. Creating more sustainable cities and conserving and restoring biodiversity are key to achieving the Sustainable Development Goals [3]. Developing cities, particularly those of the Global South, can leapfrog past unsustainable development patterns, by focusing on transformative pathways that prioritise sustainability, equity and, particularly, the socio-ecological system [4].

The global wave of unprecedented urban growth has also been recognised as one of the major drivers of biodiversity loss [5,6], with growing concern regarding the impact of

urban growth in global biodiversity priority areas [7,8]. Urbanisation in its current form will hasten the biodiversity crisis, further compromising an already strained system, and risk nonlinear and irreversible changes to the Earth system that will have a direct impact on society [9]. Africa is the most rapidly urbanising continent, where the majority are poor and approximately 60% of sub-Saharan Africa's urban population live in informal [10], often poorly serviced, settlements and are directly dependent on natural systems to meet many of their basic needs [11]. Improvement in ecosystem health through the implementation of nature-based solutions (definition as per Resolution 5 of the Fifth United Nations Environment Assembly [12]) has the potential to directly improve human well-being and is increasingly viewed as an important tool for urban areas to adapt to climate change [13].

The role of cities and local governments in addressing sustainability challenges is also receiving increasing global support and is most evident in the expression of urban-focused sustainable development goals, particularly SDG 11 of the 2030 Agenda for Sustainable Development that aims to "Make cities and human settlements inclusive, resilient and sustainable" [3]. Similarly, there is growing literature pointing to the importance of biodiversity in ensuring urban sustainability and human well-being [5,14]. Drawing these imperatives together is the responsibility of local government working with their local stakeholders, and this has been further emphasised following the adoption of the "Kunming-Montreal Global Biodiversity Framework" (GBF) by the 15th Conference of Parties to the UN Convention on Biological Diversity, which, inter alia, advocates for a "whole-of-society" approach in order to reach ambitious biodiversity targets and an emphasis on ecosystem-based approaches to minimising the impacts of climate change on biodiversity [15]. Central to this approach is increasing recognition that subnational and local governments can be the best place to drive transformative change through co-ordination of initiatives and local policy mainstreaming [16].

This call to action is particularly relevant to those cities that are within biodiversity priority areas, as failure to effectively deal with the threats of urbanisation in these areas will increase extinction rates and directly impact ambitious global biodiversity goals [7]. This is especially relevant to the 36 global biodiversity hotspots that encompass more than half of endemic plant and terrestrial vertebrate species within just 2.5% of the Earth's land surface area [17]. The City of Durban is one of 33 hotspot cities across the global biodiversity hotspots that, given their relative size and rate of expansion, have a particular responsibility in ensuring the protection of globally significant and threatened components of biodiversity [18].

Globally, biodiversity considerations are poorly integrated into urban planning, often linked to lack of supporting policy and associated planning tools [7]. Recognition of this mainstreaming gap is evident in the GBF and particularly target 12 in the urban context, which aims to "Significantly increase the area and quality and connectivity of, access to, and benefits from green and blue spaces in urban and densely populated areas sustainably, by mainstreaming the conservation and sustainable use of biodiversity, and ensure biodiversity-inclusive urban planning. . ." [15]. Similarly, the White Paper on the Conservation and Sustainable Use of South Africa's Biodiversity identifies the importance of biodiversity mainstreaming as a key enabler in achieving conservation objects and promoting sustainable development [19].

The City of Durban, with four decades of experience in the fields of urban open space planning (used interchangeably with biodiversity planning in this paper) [20], provides a useful case study in mainstreaming biodiversity through a local government lens. Over this period, there has been ongoing engagement by local government, and the lessons learnt, and approaches taken, provide useful insights for local governments of both developing and developed cities, and particularly for the network of cities that fall within biodiversity hotspots.

Therefore, we aim to demonstrate how Durban's work in open space planning has evolved over the past 40 years and to identify the key elements involved in advancing biodiversity-focused open space planning, through:

(1) The analysis of the approaches taken in response to shifting urbanisation, governance and policy/legislative contexts;

(2) Identification of the role of science and scientific information informing planning, as well as the champions required, and key institutional changes undertaken, to improve biodiversity outcomes by embedding an explicit biodiversity mandate within local government;

(3) Analysis of how biodiversity concerns have been integral in land-use planning decision-making via the city's environmental planning function (referred to as the "Department").

To address these questions, we make use of two analytical frameworks which assess the factors influencing the different open space planning iterations, and the role and influence of the Department in land-use planning processes, which is supplemented by local-level development application data.

This study serves to generate understanding, and the understanding is generalised, noting that "force of example" is underestimated in the role of case studies and the ability to generalise, which can allow for natural extension of findings beyond the boundaries of an individual study [21].

## 2. Methods

### 2.1. Location and Context

The 2566 km$^2$ City of Durban is administered by the eThekwini Municipality in the province of KwaZulu-Natal (KZN), South Africa [22] (Figure 1). The variety of landforms and climatic conditions in the eThekwini Municipal Area (EMA), as well as its unique biogeographical position, in the centre of the Maputaland-Pondoland-Albany global biodiversity hotpot [23], has resulted in a wide range of terrestrial and aquatic ecosystems that are home to a rich diversity of organisms [20]. The EMA contains three of the country's eight terrestrial biomes, viz. savanna, forest, and grassland, and includes several threatened vegetation types. In Durban alone, there are approximately 2267 plant species, 82 terrestrial mammal species and 526 species of birds. There are also 69 species of reptiles, 25 endemic invertebrates (e.g., butterflies, millipedes and snails) and 37 frog species [22].

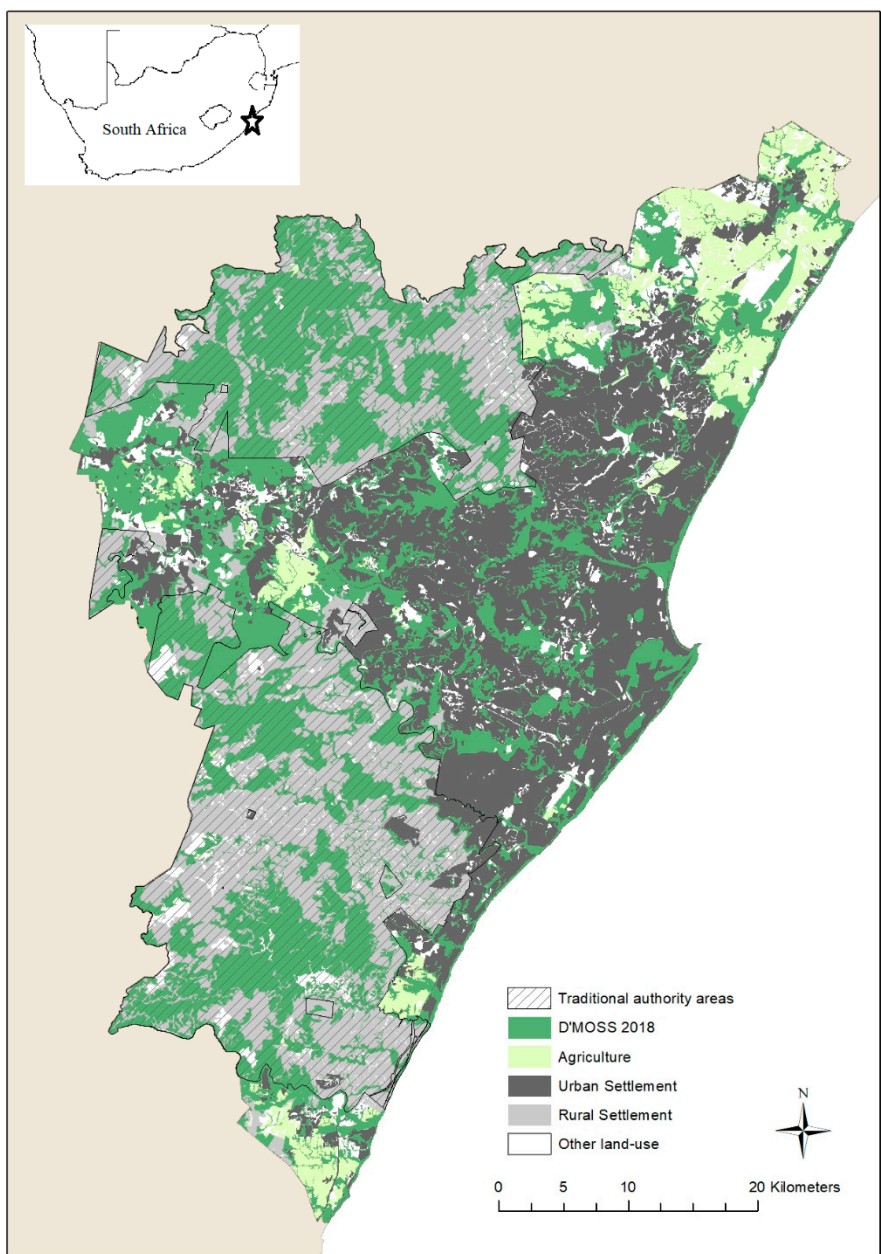

**Figure 1.** The eThekwini Municipal Area with main land uses, indicating the area administered by the Ingonyama Trust Board, with the balance falling under the formal municipal scheme and the 2018 Durban Metropolitan Open Space System (Data source: [24]).

Durban is the third largest metropolitan area in South Africa, with a population of approximately 4.1 million people [25], representing more than a third of the provincial population in an area that is less than 3% of the provincial total. The result is that Durban's rich natural resource base has been under significant pressure and negatively impacted over the past 150 years, initially by extensive agriculture, and then increasingly by rapid urbanisation, resulting in the cover of the original vegetation types being reduced to approximately one-third of the original extent [20]. Rates of loss of natural habitat are particularly high in KwaZulu-Natal, averaging 1.2% per annum between 1994 and 2011, and likely to be substantially higher in the more densely populated Durban [26].

The socio-economic context of Durban, with a high degree of inequality (Gini co-efficient = 0.62) and more than a third of the population living below the poverty line [25], further exacerbates pressure through growing informality and the increasing reliance on natural resources [24]. In addition, the availability of land to support the economic activities

associated with one of sub-Saharan Africa's busiest ports [27] is limited by, inter alia, the steep topography that characterises the city, placing additional pressure on threatened habitat types (e.g., grassland systems) that are typically associated with the flatter more "developable" areas [24]. An additional complexity in Durban is the presence of a dual governance system for the administration of land use, viz. formal municipal scheme areas that are administered by the eThekwini Municipality and traditional authority areas administered by the Ingonyama Trust Board (ITB) [24] (Figure 1). The latter, reflective of the legacy of spatial segregation, represents land held in trust for the former KwaZulu homeland area, with the Zulu King as the sole trustee [28].

### 2.2. Approach

This paper is based on 40 years of experience in the field of local government open space planning. This case study, and the associated analytical frameworks, have been used to generate understanding, and then this understanding is generalised. This approach is supported by the literature [21,29,30], with process tracing representing an important approach in providing evidence to support assertions, with theoretical implications that may extend beyond the boundaries of the case study and be comparable to large N-methodologies in generalisability [30]. The city's flagship environmental planning product, viz. the Durban Metropolitan Open Space System (D'MOSS) [31], was evaluated using an analytical framework for each of the major iterations of the product since 1982. For each iteration, the framework incorporated aspects relating to: (1) the urbanisation context of the time; (2) specific governance context that necessitated an appropriate planning response; (3) approach taken in response to contextual changes; (4) science/action nexus; (5) areas of innovation; (6) area of D'MOSS and year of approval; and (7) important champions or institutional structural changes.

A primary way in which D'MOSS is used in the municipality is through the triggering of development assessment processes linked to land-use change proposals (development applications). A second analytical framework was used to analyse the role of local government in the assessment of the environmental elements of these applications from a legislative and process context through a biodiversity lens, including: (1) applicable legislation; (2) role of the municipality in decision-making; (3) applicability of D'MOSS; (4) potential constraints to development; (5) mechanisms for mainstreaming biodiversity consideration and/or protection; (6) science/action nexus; and (7) how the biodiversity functions of the municipality contribute towards facilitation of this mainstreaming. In addition, data were collated from the Department's application database to provide the number and type of applications received per year.

Important to note is that this paper focuses on a particular aspect of Durban's response to the protection of biodiversity. There are several programmes within the city that respond to other aspects such as conservation area expansion, land management (e.g., the management of invasive alien species) and ecosystem restoration [20,32,33].

### 3. Results

#### 3.1. Durban Metropolitan Open Space System

Table 1 provides an analysis of the seven major iterations of D'MOSS using the analytical framework. The patterns are summarised below.

**Table 1.** An analysis of factors influencing the seven open space plan iterations for Durban.

| Iteration | Urbanisation Context | Governance Context That Prompted a Suitable Response | Approach Taken in Addressing the Issue | Innovation | Footprint (ha) | Science/Action Nexus | Champions and Key Institutional Structural Changes |
|---|---|---|---|---|---|---|---|
| **1982 Metropolitan Open Space System** | The Durban Functional Region comprised several local councils with limited integration of open space planning across administrative boundaries. | Concern over loss of key environmental assets, identification of imbalance in open spaces within the greater Durban area, and lack of trail systems. | Mapping of existing conservation areas, areas of conservation potential and potential trail system. | First attempt at mapping ecological assets within the greater Durban areas. | 8295 | Although there is little evidence that the map was informed by ecological theory, it represented a critical spatial product that land-use planning authorities could engage with, while also highlighting important natural spaces. | Wildlife Society (now the Wildlife and Environmental Society of Southern Africa) and the Natal Town and regional Planning Commission. |
| **1989 Durban Metropolitan Open Space System** | The municipal area at this point was restricted largely to the current CBD and suburbs immediately adjacent to the urban core. | Change in town planning legislation prompted a relook at the roles of open spaces in urban planning, with a greater focus on the role of natural areas. | Mapped network of open spaces, including nine nature reserves. | Inclusion of managed and disturbed landscapes. Economic analysis of proposed open space in terms of required capital and operational budget. | 2193 | Partnership with the university led to the development of a PhD researcher applying ecological theory in the development of an ecologically viable open space system. | Town Planning Branch, Durban; Head: Parks Department; Durban; and PhD researcher from UKZN. |
| **1999 Durban Metropolitan Open Space System Framework Plan** | New democracy and the amalgamation of 40 local municipalities substantially increased the municipal footprint. | The significant change in context placed a responsibility on government to roll out services to citizens, and a particular requirement to balance this rollout within the framework of sustainable development. | Expert-based mapping approach in the identification of sensitive areas. Recognition of open spaces as an asset that is part of the city's service delivery response. | Valuation of services delivered by open spaces and digitised using GIS software. Analysis of land included in the footprint that was considered undevelopable for reasons other than purely biodiversity. Inclusion of land under all tenure types. | 45,090 | Costanza (1997) [34] provided the theoretical framing to map and value open spaces at scale across the municipality. | PhD researcher appointed as Manager of the newly created Environmental Branch in 1994. Consultants appointed to spatially represent, and value ecosystem services provided by D'MOSS. |
| **2003 eThekwini Environmental Services Management Plan** | Further expansion of the municipal boundary, particularly the inclusion of traditional authority areas. | Increasing perception that D'MOSS represented a hindrance to the rollout of public service infrastructure. | As per phase 2, but with the removal of degraded rural and agricultural areas, and the rebranding of D'MOSS to EESMP. | Prioritisation of important areas to gain political support and move from D'MOSS to EESMP. Detailed consultation with line functions in non-environmental sectors of the municipality. | 63,115 | As per 1999; however, the inclusion of a botanist within the environmental function allowed for prioritisation of areas for inclusion in the open space network. | Development of an Environmental Management Department under the Manager of the Environmental Branch. Appointment of new staff with additional skills in the biodiversity planning and development assessment branches. |
| **2010 D'MOSS Scheme amendment** | While the area of the municipality remained unchanged since the 2003 plan, urban nodes outside of the city centre developed rapidly over this period. | Introduction of NEMBA elevated the importance of biodiversity and the need to integrate biodiversity more effectively into land-use planning. | The most comprehensive stakeholder engagement process of all the D'MOSS iterations. Approximately 18,000 letters sent to landowners as part of a land-use scheme amendment. | Inclusion into schemes as a development control layer and condition included in mapping. | 74,497 | Improved institutional capacity through the recruitment of scientists into the function allowed for the development of in-house, fine-scale land-cover data. | Incorporation of a climate adaptation function leading to a renaming of the Environmental Planning and Climate Protection Department. Appointment of a Town Planner to support the land-use scheme amendments. Substantial increase in staff under the Biodiversity Planning and Development Assessment Branches. |

Table 1. *Cont.*

| Iteration | Urbanisation Context | Governance Context That Prompted a Suitable Response | Approach Taken in Addressing the Issue | Innovation | Footprint (ha) | Science/Action Nexus | Champions and Key Institutional Structural Changes |
|---|---|---|---|---|---|---|---|
| **2016 D'MOSS** | While the area of the municipality remained unchanged since the 2003 plan, urban nodes outside of the city centre developed rapidly over this period. | The development of national and provincial vegetation maps and biodiversity plans revealed scale-related issues when applying these products at a local government scale. | Areas added to D'MOSS were because of the inclusion of critical biodiversity areas as identified, as part of a systematic conservation assessment. | Development of a fine-scale vegetation map and systematic conservation assessment for Durban. | 78,782 | The increased scientific capacity allowed for in-house development of feature data that informed the systematic conservation assessment, and additional areas, primarily based on detailed vegetation mapping, were added. Aided by a partnership with UKZN that provided additional feature data and training in conservation planning. | Manager: Biodiversity Planning and Scientists within the branch. Durban Research Action Partnership. |
| **2018 D'MOSS** | The municipal area increased in 2016 to include another traditional authority area, viz. Vulamehlo (ward 105). | Change in municipal area required an appropriate mapping response. The options-poor environment of meeting conservation targets in urban environments and promoting the protection of ecological infrastructure promoted the development of nature-based solutions under the banner of restoration ecology. | The Vulamehlo area included some of the municipality's largest and most connected natural areas, leading to a relatively large increase in D'MOSS. In addition, projects that were instated by the Department relating to the restoration of system had progressed to the point that warranted additional protection. | Inclusion of a large-scale reforestation project and future offset receiving areas. | 94,835 | Feedback loops from projects designed and implemented by scientists outside of the traditional biodiversity planning function (e.g., ecosystem-based adaptation through the Buffelsdraai Reforestation Project). | Restoration Ecology Branch responsible for undertaking the implementation of ecosystem-based adaptation projects. The Policy Branch and Environmentalists from the Biodiversity Impact Assessment Branch working in the biodiversity offset space. |

### 3.1.1. Contextual Changes and Enablers

In response to changes in urbanisation and governance contexts, the approach to open space planning has evolved over the iterations to ensure continued institutional and political support. Open space planning for the greater Durban area was originally proposed in the late 1970s by the Natal Branch of the Wildlife Society (WS), now the Wildlife and Environmental Society of Southern Africa (WESSA), whose members were concerned about the loss of important natural areas in Durban. Investigations into the role of open space planning in town planning and changes in town planning legislation, supported by a partnership between the local university and local government, provided a platform to link conservation objectives with more traditional open space planning approaches, ultimately leading to the 1989 iteration [35]. This was significant in that thus began the process of introducing science into local government planning, which would become the central tenet to subsequent iterations of D'MOSS and land-use planning in the city.

The most significant change in the urbanisation and governance context, however, was the democratisation of South Africa in 1994. The move to democracy in South Africa brought with it a host of political changes and a shift to a developmental state with a focus on addressing the issues of poverty, economic development and basic service provisioning [36]. This resulted in, inter alia, several changes to Durban's jurisdictional area that increased from 300 km$^2$ [37] of the former Durban Municipality to the amalgamation of 40 local councils in 1996 [38], inclusion of traditional authority areas in 2000 [39] and the inclusion of an additional traditional authority area in 2016, taking the total area of the eThekwini Metropolitan Municipality to 2566 km$^2$ [24].

The evolution of planning and environmental legislation emanating from South Africa's new Constitution [40] was also instrumental in providing a receptive environment for mainstreaming biodiversity-focused open space planning. The emergence of the Integrated Development Planning era, initially a result of the Local Government Transition Act Second Amendment Act (no. 97 of 1996), and the inclusion of sustainable development as a key goal of that process, meant that this responsibility was metropolitan-wide. The National Environmental Management: Biodiversity Act (Act 10 of 2004) (hereafter referred to as NEMBA) and the establishment of the South African National Biodiversity Institute (SANBI), mandated by that act for biodiversity planning and advising organs of state on biodiversity matters, represented a major step forward for the country in the mainstreaming of biodiversity into land-use planning. These key legislative advancements, therefore, provided a supportive environment for biodiversity-focused open space planning and mainstreaming. Critically, scientists were no longer external to local government during this period but were municipal employees, allowing for efficient and effective responses to mainstreaming opportunities.

### 3.1.2. Approach Taken

Approaches taken across the seven iterations have included several examples of innovation, and display of conceptual flexibility, in response to shifting urbanisation and governance contexts. For example, a key element of the 1989 iteration relative to future iterations was the intention of having all 2193 Ha of D'MOSS under municipal ownership. Importantly in the evolution of mainstreaming open space planning, and in contrast to the 1989 plan, only a quarter of the land in the 1999 D'MOSS footprint was considered public land, and the cost of securing and managing the large D'MOSS footprint through local government financing was considered prohibitive [38]. As a result, the 1999 plan placed an emphasis on management of the system through partnerships with stakeholders and, importantly for future iterations, stated the need for land-use planning tools to control future development impacts on the environment [38].

Municipal Council approval was required for all plans from 1989, and while different approaches were taken in line with shifting urban and political contexts, similarities emerge that were important in generating support, especially the role of science informing appropriate action. The first theme that emerged was the demonstration of the value of natural

assets to political leadership and citizens. The initial plans of 1982, which formed the first attempt at mapping Durban's natural assets in an open space network by the Wildlife Society (now the Wildlife and Environmental Society of Southern Africa) and the Natal Town and regional Planning Commission, and 1989, while conservation-focused, have strong links with traditional open space design most notably through, inter alia, the design of trail systems which feature prominently [35]. In the 1999 iteration, ecosystem services, and the emergence of new science in the valuation of these services through resource economics [34], provided an effective way of shifting the D'MOSS narrative from one focused exclusively on biodiversity conservation to one where it was possible to demonstrate the value of the open space system to the long-term financial sustainability of the city and its role in meeting the basic needs of poorer communities (e.g., water supply). The new D'MOSS footprint covered approximately a third of the newly formed Durban Metropolitan Area, with ecosystem services provided for by the open space system valued for the first time at ZAR 2.4 billion (then equivalent to USD 390 million) per year [38].

This conceptual shift was further advanced during the 2003 iteration and the renaming of the plan to the eThekwini Environmental Services Management Plan (EESMP). The change in name represented a deliberate move to place additional emphasis on ecosystem services, to move away from the negative public and political perceptions of a conservation-only-focused open space system that was not responsive to human needs. An important influence on this was the demarcation of a new municipal boundary in 2000 that increased the metropolitan area by 67% and included large, predominantly rural traditional authority areas [41]. The emphasis on ecosystem services was intended to provide an alignment with existing political priorities of local government leadership, who were now required to provide services to a far greater area than before.

NEMBA then provided the necessary legal rationale to put forward an argument for biodiversity protection in a developing, and increasingly options-poor, decision-making environment [22]. The identification of important areas based on targets for biodiversity features represents a key element in systematic conservation planning (SCP) [42] and an important narrative that, in combination with the legal requirements under NEMBA, was used in achieving political support and approval [22].

The second theme that emerged is the demonstration of land-use efficiency. In the 1989 D'MOSS, central to attaining Council support was an extensive economic analysis to demonstrate that the cost of acquisition of the 441 ha of privately owned land within the D'MOSS footprint, and associated operational costs, could be offset by the proposed release for sale of municipal-owned open space which fell outside of D'MOSS [35]. This represented the first example of the importance of using land-use efficiency trade-offs in gaining political support for an open space system in Durban.

In 1999, another aspect that was analysed as part of the new footprint was that approximately half of the new footprint was considered undevelopable due to physical constraints (e.g., oversteep areas or water courses) and/or underlying land-use restrictions [38]. This was carried out to mitigate concerns by city officials and politicians regarding the impact of D'MOSS on development potential. Applying the same methodology as was used in the 1999 plan to the 2003 iteration yielded a total proposed open space system of 123,000 ha, or 54% of the new municipal area [39]. Development pressure within the municipality, however, required a reduction in the total area in order to ensure long-term support for the plan [41]. In response to this, agricultural and rural settlement areas, as well as areas that had been degraded or lost through development, represented the primary land-cover attributes that were excluded from the new footprint.

Approval of the revised plan by the Municipal Council was also preceded by an extensive period of consultation with different line functions within the municipality, a requirement to receive support from Council. This was linked to the perception that the EESMP would directly impact areas available for housing projects. As a result, a detailed spatial analysis was undertaken to identify the extent of conflict between the proposed housing projects and open space system. In total, 296 planned housing project areas were

analysed against the EESMP. An overlap of only 798 Ha, or 1.2% of EESMP, was identified, which was further reduced based on the exclusion of areas that were unsuitable for housing development. The net result was the exclusion of a single proposed housing project area from the proposed EESMP and written endorsement from Metro Housing (responsible for the rollout of affordable housing projects in the city) supporting the EESMP that totalled 63,115 ha [43]. Notably, analysis of existing constraints to development within the EESMP further advanced the work first presented in 1999. Interviews with senior officials from departments with legislated spatial footprints identified areas of overlap and co-benefit (e.g., protection of flood lines and electricity servitudes) [41], further emphasising that the open space planning was promoting land-use efficiency and ensuring meaningful integration with other functions.

The targeted line function engagement approach was also adopted in the 2010, 2016 and 2018 iterations of D'MOSS (EESMP changed back to D'MOSS, see Section 3.1.3). These iterations were aided by more detailed biodiversity reporting, which illustrated, inter alia, the options-poor environment available to achieve biodiversity targets. The SCP approach that was adopted for the 2016 and 2018 iterations highlighted the inherent efficiency associated with the method of land selection [42]. Importantly, the shifts in approach were not solely concerned with the addition of areas to D'MOSS, but areas were removed due to, inter alia, loss of ecological value, outcomes from development assessments and mapping errors.

The third theme evident across the iterations was the evidence-based nature of the open space design. The 1989 plan was informed through the use of detailed ecological information in the evaluation of the municipal open spaces [44,45] and drawing on the theoretical underpinnings of Island Biogeography [46] and optimal geometric nature reserve design [47]. Future EESMP/D'MOSS iterations used predominantly desktop-based mapping approaches to characterise levels of ecological functionality and corridor identification. This shift in approach was necessary given the substantial increase in the municipal area and was made possible by use of GIS software in the development of a land-cover layer. This use of habitats as proxies for biodiversity would become a central element of subsequent iterations [22]. The development of a fine-scale land-cover map was particularly important in the evolution of the open space system. The fine scale (1:5000) of the product was fit for purpose and allowed for decisions to be taken at a cadastral level. The iterations of 1999 to 2010 were associated with improvement in the city's GIS functionality, most notably high-resolution aerial imagery produced for Durban on an annual basis by the Photogrammetry Branch, and institutionalisation of GIS skills in the Department, and within other line functions.

Mapping the city's land cover had relied on national and provincial vegetation mapping in order to assign vegetation types to the mapped land-cover units. This led to significant scale-related issues, as there were many cases in which assigned vegetation types did not match with what was present on the ground. In order to address this issue, using more detailed Durban specific data sets (e.g., fine-resolution geology and 2 m contour shapefiles), a fine-scale vegetation map was produced for Durban (Figure 2). This product would represent a key input into Durban's Systematic Conservation Assessment [22], the outputs of which would inform the substantive additions to the 2016 iteration of the D'MOSS.

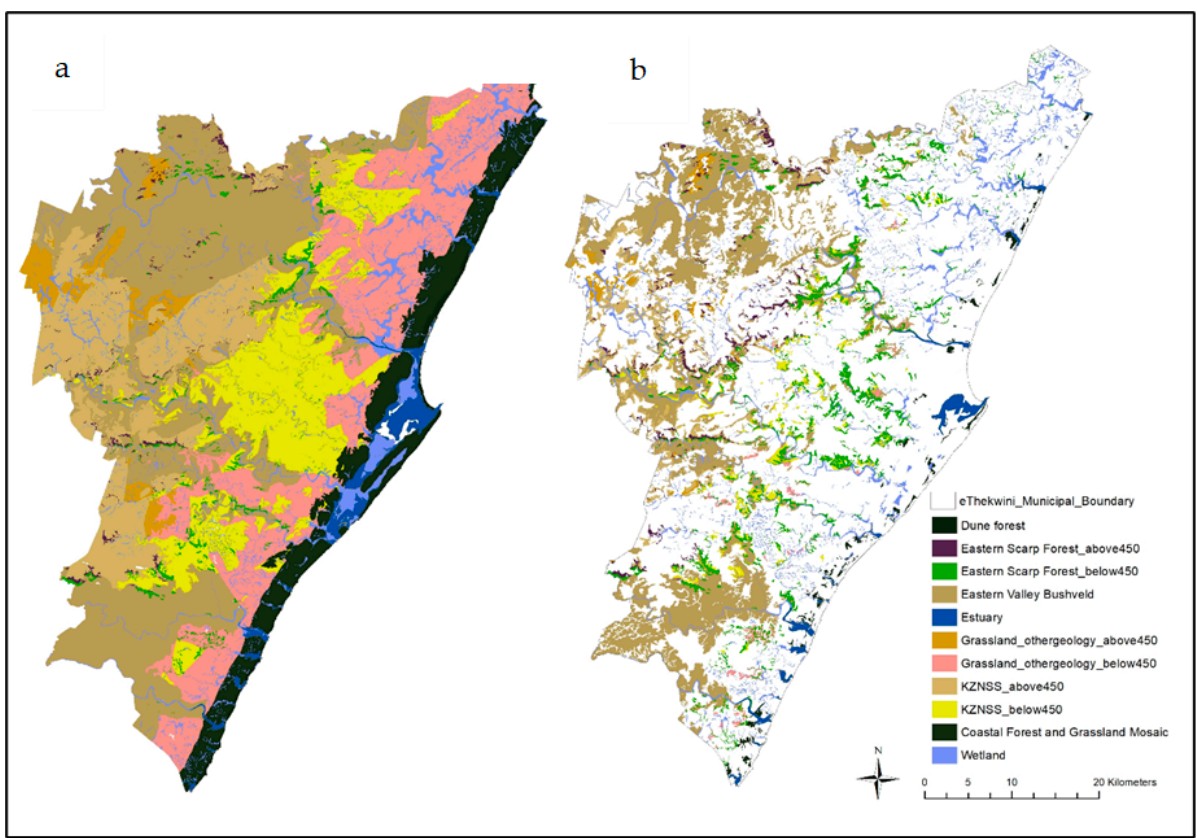

**Figure 2.** (**a**) Historical (circa 1850) and (**b**) current (2020) vegetation type maps used for municipal planning (data source: McLean 2021 [48]).

Included in the 2018 D'MOSS footprint were project areas emanating from professionals outside of the Biodiversity Planning Branch. The Buffelsdraai Reforestation Project, undertaken by the Restoration Ecology Branch, represented the output of a project that was nearly a decade in the making and was a result of the city's direct involvement in climate change science [32], as well as recognition of the importance of ecological infrastructure and D'MOSS as a climate adaptation response [33]. The project was initiated in 2008, through a partnership with the Wildlands Conservation Trust, and aimed at offsetting a portion of the carbon emissions from the 2010 FIFA$^{TM}$ World Cup through the creation of a locally indigenous forest in the buffer zone of the Buffelsdraai waste landfill [49].

The second project that influenced the 2018 D'MOSS iteration was the inclusion of offset receiving areas in the north of Durban. The project involved the development of a Sustainable Wetland Management Framework, tailored to address the contextual issues north of Durban, an area characterised by significant transformation of natural systems (largely as a result of extensive sugar cane farming) but also representing a focal point for greenfield development within the municipality. The pilot project was a partnership between two large landowners, with substantial development aspirations, and the Department, concerned with loss of remaining habitat, particularly wetland systems [50]. The project was led by the Department's Policy Branch, and environmentalists from the Biodiversity Impact Assessment Branch, and resulted in the inclusion of several degraded wetland systems that had been identified for rehabilitation.

### 3.1.3. Mainstreaming Milestones

Increasing recognition of the open space system as a service provider resulted in support from spatial planners and, ultimately, the inclusion of the layer in Durban's 2002 Spatial Development Framework [41]. This was an important development in the mainstreaming of environmental concerns within the city, ensuring that outcomes of the science-informed

plans would need to be considered in all strategic spatial planning and associated projects by other municipal functions. Durban's open space planning was, therefore, well ahead of its time in that much of the envisaged mainstreaming work (e.g., integration of biodiversity planning products into the Integrated Development Plan and Spatial Development Framework, through tools like bioregional plans and environmental management frameworks as described in NEMBA) had been in process for some time before national products emanating from NEMBA. The mainstreaming of the EESMP within the SDF allowed the Department to refocus its activities on more specialised aspects of environmental management, viz. biodiversity protection, which was not addressed in other municipal sectors and was aided by the introduction of NEMBA. This shift in focus was further emphasised by the move back from the EESMP to D'MOSS for subsequent open space plan iterations.

The key moment in the evolution of D'MOSS was the inclusion of D'MOSS into the city's Land-Use Schemes in December 2010 as a development control layer, which meant that all development applications in or immediately adjacent to D'MOSS would first need to go to the Department for approval. This was significant in that, although D'MOSS had long been included in higher-level city plans, the tension between the acknowledgement of environmental constraints based on science-informed plans and perceived development rights associated with property zoning had, to this point, not been addressed in the city's development application process. Given the extent of the D'MOSS scheme introduction, and associated public process, tension over the impact on perceived development "rights" was inevitable. While most of these objections/grievances could be addressed through focused engagements with the public, others viewed the process as illegally curtailing development rights. The matter was ultimately heard in the High Court, where a private property owner argued that the introduction of D'MOSS into land-use schemes was unconstitutional on the basis that biodiversity protection was a concurrent national and provincial legislative competence in terms of Schedule 4 of the Constitution and, therefore, not a Municipal mandate. The court, however, ruled in favour of the municipality on the basis that land-use planning is a local government competency and legislating for the environment through this process in no way impacted the mandate of other spheres of government. This judgement was momentous in that it gave the legislative mandate to the work of the Department and put an end to any discussion around local government overreaching its mandate by undertaking environmental planning [51].

### 3.1.4. Champions and Capacity Building

Institutional evolution is clearly demonstrated by the key actors involved in developing the various iterations of D'MOSS, moving from the initial work of the NGO and researchers to consultants supporting a relatively small municipal environmental function and, finally, to in-house generation within the municipality. The development of branches in the Department that cover specialised fields further entrenched institutionalisation through the creation and ownership of products (e.g., SCA and D'MOSS) and processes (e.g., development assessment), which was made possible by investment in scientific capacity in the municipal team. Scientists employed covered several fields, with a minimum qualification of an MSc. In addition, as D'MOSS increased in size and legislative influence, so did the team supporting it. This necessitated the establishment of new functions within the Department and the engagement with new areas of science, for example, in the application of restoration ecology and biodiversity offsetting concepts. Also important to institutionalisation was the inclusion of town planning professionals within the Department, which facilitated the transition of D'MOSS from policy to being entrenched in municipal legislation. The incorporation of town planners into a department dominated by environmental scientists allowed for effective implementation of biodiversity products in the land-use planning space, an established core mandate of local government [40].

Given the central role that science and application of scientific information has played in the D'MOSS story, a strong relationship with academic partners has been important. In this regard, the University of KwaZulu-Natal (previously the University of Natal) was

involved at various points across the D'MOSS iterations. The partnership leading to the 1989 iteration was the first example; however, the university was also instrumental in (1) building coastal and estuarine scientific capacity through an MSc internship programme (2007–2008); (2) contributing to the development and a departmental strategic plan (2008 and 2013); and through the establishment of the Durban Research Action Partnership (DRAP, 2011—present). DRAP uses a transdisciplinary approach to drive implementation-focused research that aims to support, inter alia, the land-use planning and management functions of the Department [52]. Research stemming from this partnership has covered several fields [20,53,54] and has been particularly important in providing biodiversity data that have informed the SCA and D'MOSS. Also key was upskilling staff in conservation planning practice through a DRAP training workshop, by leading academics from the University of Helsinki, in both the theoretical and technical aspects of conservation planning; this led to application of the methodologies in the SCA [22].

The positioning of the Department within the city's spatial planning and land-use management functions provided the ideal institutional setting to facilitate the integration of D'MOSS as a development control layer across the hierarchy of municipal plans. A key step in catalysing this change in perspective was the appointment of an Environmental Manager (the same PhD student who was instrumental in the 1989 D'MOSS plan) in the municipality in 1994, which led to the motivation for the creation of an Environmental Management Branch within the then Urban Development Department. Importantly, the same individual would remain as the head of the city's environmental function for all the subsequent D'MOSS iterations, providing leadership continuity, while proactively identifying opportunities for further mainstreaming. Examples of this have included leading Durban's local Agenda 21 programme [55] that strongly influenced the ecosystem-service-focused 1999 and 2003 iterations, shifting back to a biodiversity focus following the introduction of NEMBA [43], championing ecosystem-based adaptation [33] and resilience thinking [50]. These shifts have strongly influenced the various D'MOSS iterations.

### 3.2. D'MOSS and Development Assessment

Protection of the environment is enshrined in Section 24 (b) of the South African constitution which states that "Everyone has the right...(b) to have the environment protected, for the benefit of present and future generations, through reasonable legislative and other measures that (i) prevent pollution and ecological degradation; (ii) promote conservation; and (iii) secure ecological sustainable development and use of natural resources while promoting justifiable economic and social development." [40]. The constitution also provides for designation of responsibilities for activities and Schedule 4, part B, which assigns municipal planning responsibilities to local government [40]. These, together with various legislation, provide the mandate for Durban's and other local governments' roles, which integrate environmental planning within land-use planning processes. Table 2 provides an analysis of how the municipality engages with legislative requirements in land-use planning processes. Key elements from this table are summarised below.

Table 2. Analysis of the Department's role and influence in land-use planning processes.

| Applications | Legal Context | Legal Context (Obligation) | Legal Context (D'MOSS) | Process Context (Constraints) | Process Context (Mechanisms) | Process Context (Science/Action) | Process Context (Facilitation) |
|---|---|---|---|---|---|---|---|
| Environmental Impact Assessments | The National Environmental Management Act, 107 of 1998 (NEMA) gives effect to Section (B) of the constitution of South Africa. EIA regulations (RSA 2006, RSA 2010, RSA 2014, RSA 2017) and associated listing notices. | The eThekwini Municipality is a commenting authority in terms of NEMA processes within Durban. | Flagging layer in which applicants are advised of possible triggers for EIAs | Threatened habitat types and critical biodiversity areas as identified in Durban's Systematic Conservation Assessment and Durban Metropolitan Open Space System. | Comments from staff (registered environmental professionals) relating to the protection of the natural assets that may include:<br>• The need for additional specialist studies.<br>• Changes in development layout.<br>• Motivation for the protection of the features based on fine-scale information. | Scientists employed to:<br>1. Develop fine-scale biodiversity data that are used to inform the need for an assessment, and to inform comments provided during the process.<br>2. Provide specialist input in relation to the review of submissions (e.g., wetland specialist). | • Departmental Enquiry form.<br>• provision of additional biodiversity feature data to applicants, Environmental Assessment Practitioners, and specialists. |
| Category 1 land development determinations (e.g., introduction and amendment to land-use schemes) | The Spatial Planning and Land Use Management Act, 16 of 2013 (SPLUMA) and the eThekwini Municipality's Planning and Land-Use Management By-Laws, 2016 (Chapter 8, Section 26). | The eThekwini Municipality is the competent authority in terms of land development applications. The decision-maker for applications falling within this category is City Council. | As D'MOSS is part of the SDF, it represents a key informant in the development and introduction of land-use schemes. The Department is a key contributor function in the development of the package of plans. | • Current and future conservation areas.<br>• Areas with threatened vegetation types.<br>• Critical biodiversity areas.<br>• Other important features (e.g., buffer areas, ecological corridors, ecosystem service areas). | • Comments provided by the Department on the various packages of plans.<br>• Refinement of the D'MOSS layer.<br>• Where appropriate, the identification of parcels for additional environmental protection.<br>• Additional input provided through the Joint Advisory Committee through the Department's in-house professional planner. | Scientists employed to:<br>1. Develop fine-scale biodiversity data that are used to inform the development of plans.<br>2. Review work undertaken by consultants. | • Terms of reference for the use and integration of environmental data for consultants within the package of plans.<br>• Provision of biodiversity data.<br>• Field verification by biodiversity specialists. |
| Municipal Planning: Category 2: Departures from the SDF, zoning and rezoning of land. | The Spatial Planning and Land Use Management Act, 16 of 2013 (SPLUMA) and the eThekwini Municipality's Planning and Land-Use Management By-Laws, 2016 (Chapter 8, Section 27). | The Municipal Planning Tribunal (MPT) is responsible for making decisions on this category of applications. The MPT is comprised of designated municipal officials and persons appoint by the City Council with extensive experience with, inter alia, spatial planning. | The MPT Is governed by SPLUMA and the bylaws and must consider the environment within decision-making. | • Current conservation areas.<br>• Areas with threatened vegetation types.<br>• Critical biodiversity areas,<br>• Other important features (e.g., buffer areas, ecological corridors, ecosystem service areas). | • Comments from registered environmental professionals indicating whether the application is supported or not, and/or supported with conditions of approval (e.g., protection of remaining biodiversity features.<br>• Registered environmental professional forms part of the MPT and ensures oversight of environmental impact.<br>• Additional input provided through the Joint Advisory Committee through the Department's in-house professional planner. | Scientists employed to:<br>1. Develop fine-scale biodiversity data that are used to inform comments.<br>2. Provide specialist comments (e.g., wetland and botanical specialists). | • Enquiry form.<br>• Provision of biodiversity data.<br>• Field verification by biodiversity specialists. |

**Table 2.** *Cont.*

| Applications | Legal Context | Legal Context (Obligation) | Legal Context (D'MOSS) | Process Context (Constraints) | Process Context (Mechanisms) | Process Context (Science/Action) | Process Context (Facilitation) |
|---|---|---|---|---|---|---|---|
| Municipal Planning: Category 3: Special consent applications, applications for subdivision, and development of land outside of a land-use scheme. | The Spatial Planning and Land Use Management Act, 16 of 2013 (SPLUMA) and the eThekwini Municipality's Planning and Land-Use Management By-Laws, 2016 (Chapter 8, Section 28). | These applications are considered and decided by the Head: Development Planning, Environment and Management. | By virtue of inclusion in land-use schemes, all applications received by the land-use management office will be allocated to the Department for review and consideration. Applications that fall outside of the scheme will also be referred to the Department as D'MOSS is a foundational element of the SDF. | • Threatened vegetation types. <br>• Critical biodiversity areas. <br>• Other important features (e.g., buffer areas, ecological corridors, ecosystem service areas). | • Comments from registered environmental professionals indicating whether the application is supported or not, and/or supported with conditions of approval (e.g., protection of remaining biodiversity features). <br>• Additional input provided through the Joint Advisory Committee through the Department's in-house professional planner. | Scientists employed to: <br>1. Develop fine-scale biodiversity data that are used to inform comments. <br>2. Provide specialist comments (e.g., wetland and botanical specialists). | • Enquiry form. <br>• Provision of biodiversity data to applicants and consultants. <br>• Field verification by biodiversity specialists. |
| Municipal Planning: Category 4: relaxation and exemptions from the provisions of the land-use scheme. | The Spatial Planning and Land Use Management Act, 16 of 2013 (SPLUMA) and the eThekwini Municipality's Planning and Land-Use Management By-Laws, 2016 (Chapter 8, Section 29). | These applications are considered and decided by the Deputy Head: Development Planning. | By virtue of inclusion in land-use schemes, all applications received by the land-use management office will be allocated to the Department for review and consideration. | • Threatened vegetation types. <br>• Critical biodiversity areas. <br>• Other important features (e.g., buffer areas, ecological corridors, ecosystem service areas). | • Comments from registered environmental professionals indicating whether the application is supported or not, and/or supported with conditions of approval. | Scientists employed to: <br>1. Develop fine-scale biodiversity data that are used to inform comments. <br>2. Provide specialist comments (e.g., wetland and botanical specialists). | • Enquiry form. <br>• Provision of data to applicants and consultants. <br>• Field verification by biodiversity specialists. |

### 3.2.1. Local Government Role in the National Environmental Management Act: EIA Regulations

The National Environmental Management Act (Act 107 of 1998) gives effect to Section 24 of the constitution, and, while competency is assigned to national and provincial environmental authorities, NEMA obliges all organs of state to align with the principles of the Act (see Fuel Retailers Association of Southern Africa v Director-General: Environmental Management, Department of Agriculture, Conservation and Environment, Mpumalanga Province & others, 2007 [56]). The municipality and Department play a key role in ensuring compliance with this legislation, and the associated EIA regulations, through development assessment processes. The Spatial Planning and Land Use Management Act (Act 16 of 2013) and eThekwini Planning By-Law (13 of 2016) give effect to the constitutional mandate of the municipality in the land-use planning mandate. Application for land-use changes in terms of the eThekwini By-Law are divided into two stages, viz. pre-application enquiry and formal lodgement.

By virtue of the inclusion of D'MOSS within the land-use scheme in 2010, all enquires that potentially impact D'MOSS are directed to the Department for comment. The initiation of an enquiry process has been particularly important in that it provides landowners with clear direction on what process to follow, thereby reducing unnecessary financial outlay and preventing unrealistic development aspirations. Included within the correspondence is the identification of potential triggers that may require additional processes (e.g., Basic Assessment or EIA) in terms of NEMA, as well as an indication of whether the Department would support the application during the NEMA process.

Not only is there an important role in the identification of "triggers", but the fine-scale nature of the city's biodiversity data, particularly when it comes to the mapping of ecosystems, can equally drive the need for NEMA processes. This is particularly relevant for the distribution of threatened ecosystems, which were first gazetted in 2011 [57], with the intention of ensuring that any potential impacts to critical biodiversity would be assessed at lower "trigger" thresholds than ecosystems under less threat. As an example, in terms of activity 12 of listing notice III of NEMA EIA regulations for the province of KwaZulu-Natal, the clearance of 300 $m^2$ of endangered or critically endangered vegetation types requires at least a Basic Assessment. The equivalent "trigger" threshold for clearance of indigenous vegetation that is not listed as threatened is 10,000 $m^2$. This type of activity is frequently encountered within Durban, and given the coarse nature of the national and provincial vegetation maps, many of these impacts would proceed without assessment. The finer-scale vegetation mapping and associated D'MOSS development application processes do compensate for these shortcomings by alerting landowners to the presence of threatened ecosystems at a useful spatial scale. This scenario is commonly associated with the distribution of the Endangered KwaZulu-Natal Sandstone Sourveld Grassland that the Department has been able to more accurately map within Durban [20]. These data are provided to landowners and environmental consultants acting on behalf of the applicant.

### 3.2.2. Municipal Planning and D'MOSS

In contrast to the commenting role within NEMA processes, the municipality is the decision-maker (mandated authority) in terms of land-use planning applications. SPLUMA (Act 16 of 2013) provides the primary legislative framework for spatial planning within the country and, inter alia, requires municipalities to: (1) compile Integrated Development Plans and an associated Spatial Development Framework; and (2) prepare and implement a Municipal Planning By-Law, including the preparation of land-use schemes, that act to control and regulate the use of land within the municipal area [25]. In response to the latter, the eThekwini Municipality Planning and Land Use Management By-Law (13 of 2016) was gazetted in August 2017. The bylaw provides guidance in terms of how different categories of land-use applications are processed within the municipality.

Table 2 provides an analysis of the different categories of land-use applications and how biodiversity information is integrated into decision-making. Across all categories,

there is evidence of active participation in land-use planning processes from the more strategic municipal-led category 1 applications to the predominantly private land-use applications associated with categories 2–4. Across all applications, an emphasis is placed on ensuring that fine-scale biodiversity information is applied, whether this be through the provision of fine-scale data to applicants or, where appropriate, site assessments. An example of this was the development of the Cato Ridge Local Area Plan and Draft Scheme [58], a category 1 application, and an area that had been identified for significant industrial development, but representing a centre for remnant parcels of threatened grassland systems. Through detailed engagement with the process, professionals from the Department were able to motivate, through refinement of the fine-scale vegetation mapping, for selected, previously unzoned sites to receive a proposed land use and zoning of "conservation" [58].

Mainstreaming of biodiversity into land-use decision-making has also been advanced through capacity development. Comments relating to applications for potential development in D'MOSS areas were undertaken by professionally registered environmental scientists who could interpret biodiversity information and potential impacts and provide informed recommendations for decision-making. Decisions include: (1) Approval without conditions—as is the case with areas that, following assessment, are found to be highly degraded, there has been a mapping error or in cases where there is no likely impact on D'MOSS. (2) Approval with conditions—these can include: requirements to change the development layout to avoid biodiversity features and associated buffers; the protection of remaining biodiversity features on the affected site through an appropriate protection mechanism (e.g., conservation servitude registered in favour of the municipality against the property title deed or conservation zone), thereby adding to the network of conservation areas in the city [59]; and/or biodiversity offsets. Biodiversity offsets are, however, considered a last resort and typically occur following a NEMA process relating to the loss of threatened habitats. (3) Not approved/supported—the loss of threatened vegetation types is not supported by the Department [43], which may lead to an application not being approved. The subdivision of land where the new proposed land parcel contains threatened vegetation is an example of an application that is typically not approved. Under exceptional circumstances, when applications for important vacant sites are received, the Department may approach the landowner to purchase the land to prevent potential loss and add to the conservation areas network. This was, however, considered a last resort given the relatively small land acquisition budget for the Department [59].

Capacity development in key institutional structures has also been important in advancing mainstreaming. The first example related to the inclusion of a registered environmental professional to serve on the Municipal Planning Tribunal that deals with, inter alia, land-use zoning, and given the potential risk of this category of land-use planning application to biodiversity, represents an important example of the institutionalisation in ensuring that biodiversity concerns are considered in this application category. The second example related to the presence of a registered professional town planner within the Joint Advisory Committee that provides recommendations on applications across categories 1–3. This appointment was critical in guiding proactive processes (e.g., the approval of D'MOSS), advancing biodiversity mainstreaming across spatial planning processes, and ensuring that comments relating to biodiversity impacts have been adequately considered.

3.2.3. Applications Received by the Department

Figure 3 shows the breakdown of development applications received by the EPCPD over an eleven-year period, totalling approximately 17,000 applications. Over the course of this period, only 15% of the applications received were the result of NEMA Environmental Impact Assessment (EIA) applications. The remaining applications were comprised of building plans (33%), enquires (31%), and other planning applications (e.g., rezoning applications, 21%), including municipal infrastructure projects.

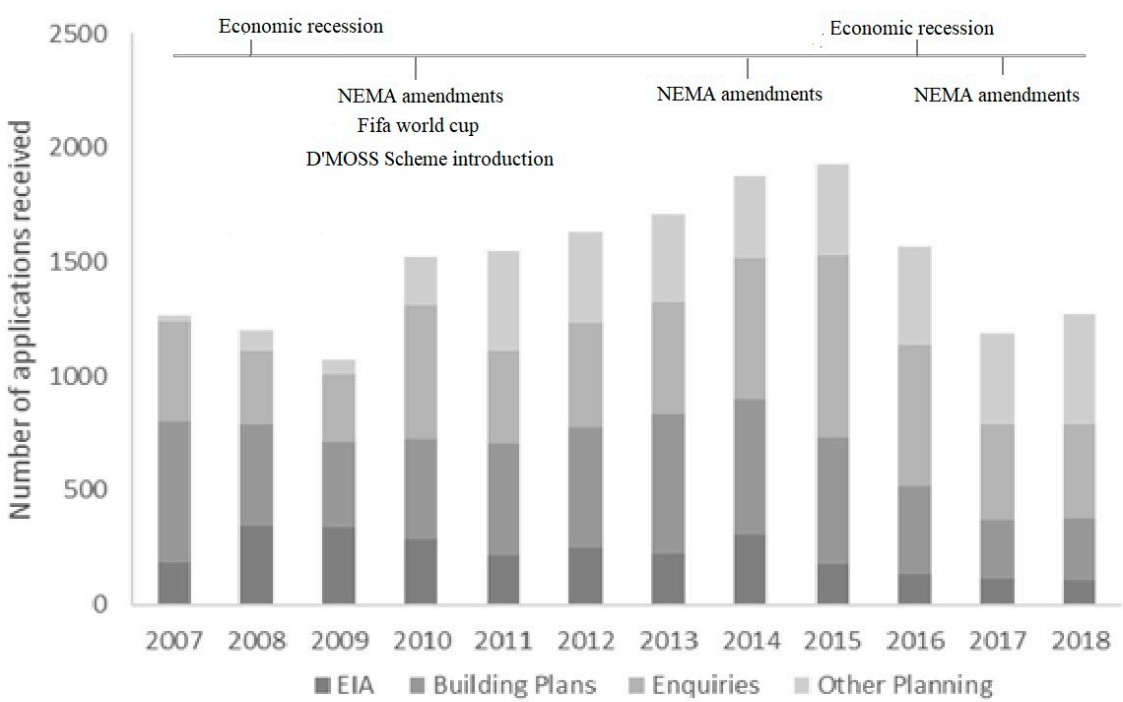

**Figure 3.** The number of development applications received by the Department from 2007 to 2018.

The number of applications received mirrors prevailing economic conditions, indicated by the lower numbers recorded during the economic recessions of 2008/2009 and 2016/2017. Even though the total number of applications received were comparatively low over 2008 and 2009, the impact of the 2010 FIFA World Cup$^{TM}$ is evident in the number of EIAs received in the years preceding this, which were the highest over the eleven-year period.

Policy and legislative changes also influenced the overall numbers. The adoption of D'MOSS as part of the town planning schemes, for instance, played a role in the increase from 2009 to 2010 in the total number of applications, particularly with the number of enquiries and other planning applications received over the respective years. The number of EIAs received were also directly related to changes in NEMA regulations and, in particular, changes to the thresholds for "triggering" NEMA processes. Major changes to the regulations took place in 2010 and 2014, and in both instances, the number of EIAs received decreased, which may be related to relaxations of thresholds for certain listed activities (e.g., road construction) and the exemption of certain listed activities (e.g., pipelines transporting water, sewage and storm water) within urban areas [60]. The inclusion of D'MOSS within schemes in 2010 has provided a necessary safety net in assessing impacts that no longer meet thresholds triggering NEMA processes and ensuring that important biodiversity features were not being compromised by "death by a thousand cuts" as a result of the cumulative impact of small-scale development applications [20].

## 4. Discussion

This paper documents the evolution of Durban's open space planning over the past 40 years and contributes to addressing the paucity of research in this field from countries of the Global South [61,62]. Through the analytical frameworks, there is evidence of the advancement of mainstreaming biodiversity concerns within local government processes, institutional functions and land-use decision-making. This has been achieved through effective and sustained leadership; the use of science and scientific information in advancing the policy and legislative environment and building political support by responding to shifting governance contexts; investment in institutional scientific capacity, and generating scale-appropriate biodiversity information (Figure 4). These are areas that are typically seen as barriers to implementing urban biodiversity planning [63], and where

Durban's biodiversity function has invested sustained resources and displayed innovation in addressing challenges.

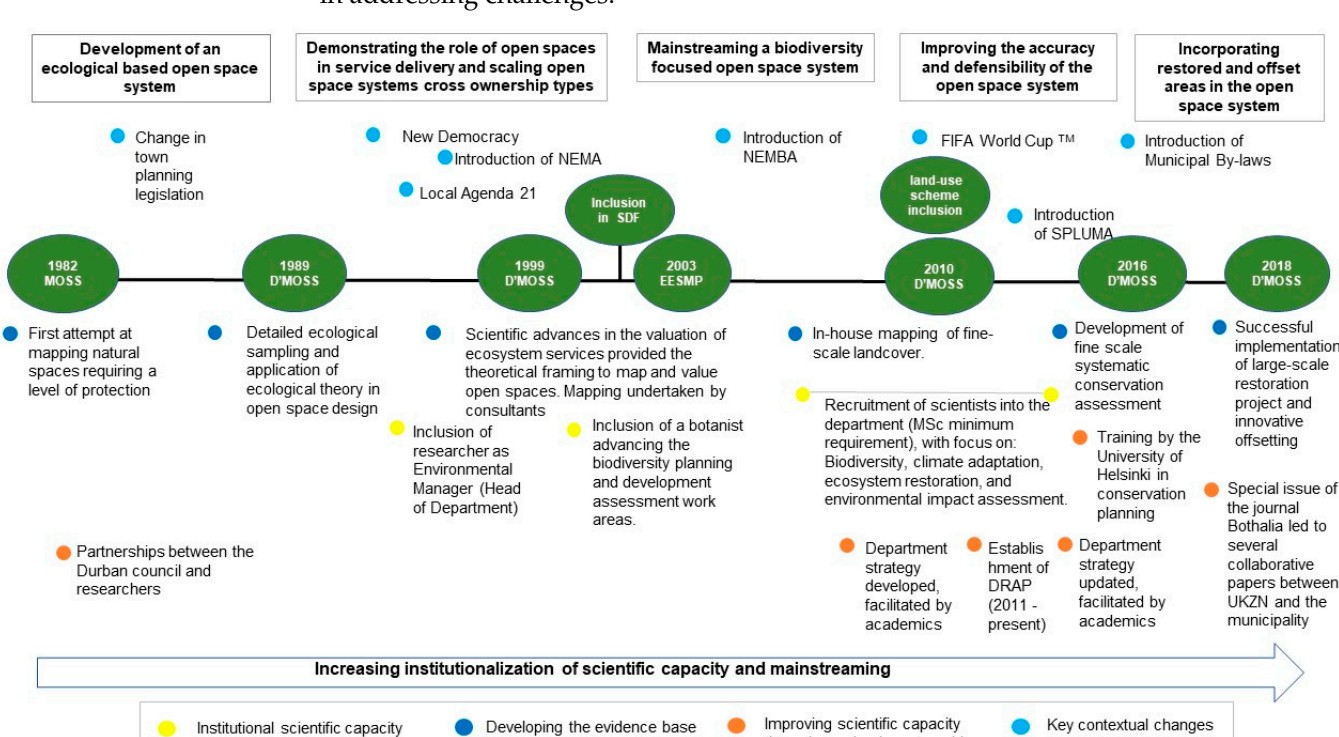

**Figure 4.** The role of science in the evolution of open space planning in Durban (see Section 3 for details).

It took almost three decades from the origin of D'MOSS to have it fully embedded within town planning schemes, supporting the view that the time-consuming nature of mainstreaming biodiversity and land-use policy implementation are often not fully appreciated [61,64]. The number of D'MOSS iterations produced during this period further emphasised that effective mainstreaming cannot be achieved through generation of a single product [65] but requires that policy capacity be maintained and evolve over time [66]. This responsibility often falls to individuals/departments to champion the biodiversity cause at a local level [67]. While several champions and key actors have emerged through the different iterations of open space plans, the role of the researcher who contributed to the 1989 iteration of D'MOSS and subsequent Head of Department has been important in driving the strategic direction of the function, elevating Durban's environmental profile [31], ensuring the preservation of institutional memory and providing leadership continuity.

An important component of local government leadership/champions is the ability to identify opportunities for policy advancement and linking to the political agenda to ensure continuity [63]. This has been clearly demonstrated in the "conceptual flexibility" applied to the shifting social, economic and political pressures. This "conceptual flexibility" would emerge as a constant through the evolution of open space planning in Durban. Drawing on findings from another city, the City of Melbourne has demonstrated significant policy advancement in mainstreaming biodiversity concerns through the intentional transition from an "Urban Forestry" to an "Urban Nature" policy agenda [68]. These shifts in focus reflect the changing global discourse of the time, while supporting the premise that conservation actions, and particularly in an urban setting, operate within a social-ecological system that requires continual adaptation to changing contexts [69]. In the case of Durban, these shifts in focus were an intentional move to reinforce the open space system as an asset directly tied to service delivery, which is often not adequately recognised by developing cities [70]. Furthermore, consistent messaging aligned with prevalent global environmental discourse of the time served to educate officials and politicians, thereby creating familiarity

with concepts that could otherwise be a barrier to implementation [71]. An example of this is the recognition of D'MOSS as a key climate adaptation response within Durban's Climate Change Strategy [33].

A common theme that emerged across the open space planning iterations and was clearly used to foster institutional and political support was the ability to demonstrate land-use efficiency in open space plan design. This was achieved through: (1) trade-offs in municipal-owned land parcels in the 1989 iteration that was able to demonstrate financial sustainability of the proposed plan. In a resource-constrained environment, land trading to maximise biodiversity conservation is often inevitable [72], and understanding the full suite of costs in the design of a habitat network is essential to ensure cost-effective allocation of resources for biodiversity conservation [73]. (2) The quantification of areas within the open space plans considered to be undevelopable was important in demonstrating efficiency when open space planning shifted from an envisaged network of municipal-owned land to a development control layer across ownership types [35,38]. These types of synergistic land-use approaches are prevalent in ecosystem-based adaptation approaches to disaster risk reduction [74], and the identification of co-benefits helps build support amongst stakeholders and decision-makers [75]. These are areas that also represent opportunities for future conservation area expansion [59]. (3) Finally, the adoption of systematic conservation planning as a primary methodology informing open space design provided an important communication tool within the city, as land-use efficiency (or complementarity) is a central component of the method [42], which had been adopted and applied extensively within the conservation agencies of the country [57].

The lack of institutional capacity represents the most cited barrier to mainstreaming biodiversity considerations into land-use planning [7,14,61,65], and particularly in the development of techno-scientific knowledge and the ability to apply this knowledge within the policy landscape [63]. Central to the success of environmental policy at the local level is that it is grounded in science [76], as demonstrated in the City of Cape Town [77]. The evidence-based nature of Durban's open space plan design, made possible by the recruitment of technically capable staff with scientific training, has contributed towards gaining approval, institutionalisation of biodiversity concerns, and integration within development processes. The role of academics, and more specifically the University of KwaZulu-Natal, has also been important at various points in assisting in scientific capacity development, knowledge generation and ensuring that strategic direction is in line with the global thinking of the time [52].

Importantly, as a result of the improved institutional scientific capacity, ecosystem mapping has been undertaken at a scale that is appropriate for urban planning (i.e., relevant at the individual property level) [78]. This is seemingly an underappreciated aspect of urban conservation planning but, as has been highlighted in the case of threatened grassland system [20], represents an important innovation towards achieving biodiversity targets and preventing loss through development assessment processes. Similarly, in a review focusing on the use of biodiversity data in spatial planning and impact assessment within the European Union, scale of biodiversity data supplied through online data portals was highlighted as a challenge for practitioners operating in this field [79]. There are cities that have a long history of ecological research that has informed conservation planning [77,80,81]. These cases represent outliers, as most cities that have local-level biodiversity data are reliant on rapid and cost-effective habitat mapping as surrogates for biodiversity [67,82].

We certainly support the use of habitat maps as surrogates for biodiversity as an interim approach for cities with limited resources as it, inter alia, provides a data foundation that can immediately interact with urban spatial planning and land-use management. The caveat for this approach, however, is that habitat maps can be poor surrogates for certain species [83] and taxonomic groups [84], and require interrogation and adaptation over time [72]. Effective sustainable environmental decision-making requires investment in appropriate resolution environmental data [79].

The greater focus on implementation of conservation plans in South Africa has resulted in increasing recognition of the importance of fine-scale plans [85], which is a welcome response in an attempt to addressing aspects of the "implementation gap" that is prevalent in the conservation planning field [86]. Key to closing this gap has been the evolving national legal framework (see NEMA, NEMBA and SPLUMA) that has advanced biodiversity mainstreaming at the local level, compelling local-level political support and moving biodiversity planning at the local level from a voluntary endeavour to an auditable legal requirement. Based on experiences from Durban, key spatial outputs such as the national vegetation maps [87] that are linked directly to land-use legislation [57] have advanced biodiversity mainstreaming but require adaptation to ensure effectiveness at the local level [88]. Adaptive design in conservation planning requires both scaling-down and scaling-up processes [89]. This paper has highlighted the value of fine-scale data in land-use planning processes, and given the significant impact that listing of threatened ecosystems has had on development assessment [57], there is a risk that coarse data will lead to poor biodiversity outcomes. Focused attention on accurate delineation of threatened ecosystems is required, or whether a more aggregated approach to vegetation classification in certain biomes/regions could promote a better biodiversity outcome [90]. The second recommendation relates to "scaling up", as there is a clear need for the creation of a platform for local governments to engage with mandated provincial and national conservation authorities to effectively integrate fine-scale data into national products, while acknowledging the constraints associated with balancing standardisation with innovation [85]. This iterative and responsive approach represents a key component of transitioning from the regional scale to local action [89] and supporting vertical integration across the spheres of government that is essential for effective land governance [19,91].

Building capacity in the application of biodiversity information in the context of land-use planning processes by local government officials is also important as decisions can have a major impact on biodiversity outcomes [64]. This has been advanced by the in-house generation of biodiversity data and conservation planning products, thereby strengthening the link between conservation planning and implementation [92]. An important intervention, however, was the recruitment of individuals with training in environmental sciences who were able to use this information within development assessment processes. In a study of six municipalities in the Eastern Cape (South Africa), capacity constraints in small and intermediate municipalities highlighted the risk of EIA requirements for certain projects going unnoticed [93]. The development of screening processes and active engagement in the development assessment space, as demonstrated in this paper, has contributed to slowing the rate of biodiversity loss in Durban and, in some cases, even contributed to active protection of sites [59]. Measuring the impact of conservation practises, however, is challenging [94], and especially so in measuring avoided loss [65]. Despite these challenges, further research is required to advance our understanding of the impact these professionals have in reducing biodiversity loss in Durban. These data are essential for identifying areas of future focus, including the need for the refinement of existing tools or the creation of new ones. In this regard, and given the ongoing pressure for large-scale development, an offset policy for Durban needs to be considered to address inconsistencies in adherence to established offset principles [95,96].

Addressing the planning–implementation gap in conservation planning field [86] requires, inter alia, a receptive institutional setting and associated tools for integrating conservation planning products [97]. Institutional positioning of the Department under the Unit responsible for spatial planning and land-use management has allowed for greater capacity development, policy and legislative advancement, and mainstreaming of biodiversity concerns. An example was the inclusion of a town planner within a biodiversity-focused department, who was able to navigate planning regulations and process requirements in the transition of D'MOSS from policy to legislation, and supporting the position that multidisciplinary approaches are essential in achieving biodiversity conservation outcomes [98]. In a comparison between the approaches taken between Cape Town and

Durban in conservation plan implementation, the institutional positioning of Durban's environmental function under the Spatial Planning Division was identified as a key factor in the effectiveness of land-use planning interventions [80]. As demonstrated in this paper, the importance of institutional positioning, however, extends beyond land-use integration, and includes access to, and influence on, land-use decision-making processes. This type of horizontal integration is vital to ensure continued mainstreaming of biodiversity within decision-making and an essential component of effective governance within a dynamic socio-ecological system [99]. In a comparison between intermediate-sized and metropolitan municipalities, it was shown that, despite conservation maps being present within the SDFs of intermediate-sized municipalities, land-use planners did not consider them in routine work and environmental officers present in the same organisation were typically not included [93]. These types of inconsistencies are prevalent across local governments in South Africa and other countries [63,100,101] and represent a significant threat to biodiversity and envisaged mainstreaming. Identifying opportunities to mitigate inconsistencies is important, as has been highlighted by the inclusion of a registered environmental professional as part of the Municipal Planning Tribunal, which represents a minor but potentially influential governance response.

In conclusion, this paper demonstrates the important role a local government can play in addressing the biodiversity crisis through land-use planning and that effective mainstreaming requires considerable, focused effort and resourcing that needs to be sustained over time. This case study may be valuable to other local governments, particularly those situated in similar socio-ecological contexts. It is hoped that case studies like these, and the lessons learnt, will promote more rapid and effective mainstreaming in other local governments. Science has been a central actor in the evolution of D'MOSS and the responsiveness to contextual changes, and there is a need for this to remain the case, especially in the face of a changing climate and urbanisation pressures [102]. Ecological processes will be altered, and greater reliance on natural assets to mitigate extreme natural events will be required [103]. Given the complexities of this urban-biodiversity nexus, the transdisciplinary nature of the DRAP programme represents an important vehicle to identify the next conceptual shifts and ensure that the D'MOSS remains fit for purpose.

**Author Contributions:** Conceptualization, C.T.M., D.C.R. and R.S.; Methodology, C.T.M., D.C.R. and R.S.; Formal analysis, C.T.M.; Investigation, C.T.M.; Writing—original draft, C.T.M.; Writing—review & editing, C.T.M., D.C.R. and R.S.; Supervision, D.C.R. and R.S. All authors have read and agreed to the published version of the manuscript.

**Funding:** This research received no external funding.

**Institutional Review Board Statement:** Not applicable.

**Informed Consent Statement:** Not applicable.

**Data Availability Statement:** The original contributions presented in the study are included in the article, further inquiries can be directed to the corresponding author.

**Acknowledgments:** This study was supported by eThekwini Municipality through the Durban Research Action Partnership.

**Conflicts of Interest:** Cameron McLean is currently employed with the eThekwini Municipality. Debra Roberts was employed at the eThekwini Municipality up until her retirement at the end of January 2024. The remaining author declares that the research was conducted in the absence of any commercial or financial relationships that could be construed as a potential conflict of interest.

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
