# Peer review of "The Evolution of Open Space Planning within a Developing, Biodiverse City (Durban, South Africa)"

_sustainability, doi:10.3390/su16073073_

Round 1

Reviewer 1 Report

Comments and Suggestions for Authors

Well written payer, results are clearly presented. Authors know the context in detail and they provided a model that takes into account local and global parameters.

I dont have anything to add, excep a minor issue regarding the Figure 4, p. 18. It could be organized better (we dont see well the interesting details, it might be organized in a different way... in two sections?)

Reviewer 2 Report

Comments and Suggestions for Authors

Dear Author/s,

at the beginning, I would like to emphasize that the presented article is valuable and worth publishing. The topic discussed, i.e. biodiverse cty, is topical. The Introduction to the article is interesting - it justifies the need to conduct research. The aim formulated in the Introduction concerns subsequent tasks undertaken by the Author/s. My suggestion, The Introduction should be supplemented with a description of the structure of the text - the reader should know what will be analyzed in the following parts. A description of the text structure should be placed in the last paragraph of the Introduction.

 Although the article is interesting and valuable, I would like to emphasize that the article is a typical case study, so it belongs to the group of empirical articles, its contribution to science is rather average.

Secondly, the article completely omitted the description of the research method. The section entitled Method describes only the city of Durban as the subject of the research and research tasks.
In my opinion, the Author/s used the case study as a scientific study and although its usefulness for theoretical generalizations is limited, it is an excellent tool for formulating hypotheses - which, in my opinion, the Author/s did. In my opinion, you (authors) should describe the research method used and the research scheme - the research has its own systematics, which should be described in detail by you (authors).
The description of the tasks performed is detailed and clear, and the discussions conducted by the authors are clear. One suggestion: in my opinion, the importance of the scientist as a factor influencing the local government's attitude towards changes in biodiverse cty is only overemphasized. Based on the case of the city of Durban, one can only hypothesize that the introduction of scientific research at the local level may lead to an improvement in the biodiversity situation. Certainly, a categorical assessment of the factors cannot be made based on this case, as is done in the article.
Lastly, I find it incomprehensible to compare the Durban case with the situation in Melborne or Cape Town.

To sum up, below are my synthetic comments on the text:

In the ntroduction part, you (Author/s) clearly presented the issue of biodiversity in the area of the city of Durban. You (Author/s) justified the need to include biodiversity in regulations at the institutional level (government and local). The introduction lacks a description of the structure, which should be described in the last paragraph of the introduction. Without this, the reader does not really know what to expect from the content.

The research methodology is described incompletely. The text lacks a description of the research methods and procedures used - the lack of this significantly affects the value of the text.

 The empirical value of the article, conclusions and implications described in detail refer to the issues discussed in the article. It would be worth softening the conclusions, because it is difficult to draw such specific conclusions based on the analysis of only the city of Durban.

 I assess the whole text as a valuable text.

Kind regards,

Reviewer 3 Report

Comments and Suggestions for Authors

This is a very interesting city level study which represents a useful addition to the relevant urban global south literature. That said there are some areas that would be useful to have further commentary.

1) how are issues of alien invasives to be best managed, both in the present and in the future. Given the very significant trading role of the city

2) perhaps in part related to the above the city has a range of exotic species already present, some of which may nevertheless be significant with respect to biodiversity with many of them being naturalised of course. What specific strategies are being developed with respect to their maintenance or gradual replacement?

3) please comment in more detail about how the city's biodiversity strategies are integrated into other aspects of urban planning, particularly with respect to recreation, tourism, and climate change adaptation

Comments on the Quality of English Language

minor editing only required
